# HaplotagLR: An efficient and configurable utility for haplotagging long reads

**Monica J. Holmes[1], Babak Mahjour[2], Christopher P. Castro[1], Gregory A. Farnum[1], Adam G. Diehl[1], Alan P. Boyle**[1,3]*

**1** Department of Computational Medicine and Bioinformatics, University of Michigan, Ann Arbor, Michigan, United States of America, **2** Department of Medicinal Chemistry, University of Michigan, Ann Arbor, Michigan, United States of America, **3** Department of Human Genetics, University of Michigan, Ann Arbor, Michigan, United States of America

* apboyle@umich.edu

**Data Availability Statement:** All relevant data are within the manuscript and its Supporting Information files.

## Abstract

Understanding the functional effects of sequence variation is crucial in genomics. Individual human genomes contain millions of variants that contribute to phenotypic variability and disease risks at the population level. Because variants rarely act in isolation, we must consider potential interactions of neighboring variants to accurately predict functional effects. We can accomplish this using haplotagging, which matches sequencing reads to their parental haplotypes using alleles observed at known heterozygous variants. However, few published tools for haplotagging exist and these share several technical and usability-related shortcomings that limit applicability, in particular a lack of insight or control over error rates, and lack of key metrics on the underlying sources of haplotagging error. Here we present HaplotagLR: a user-friendly tool that haplotags long sequencing reads based on a multinomial model and existing phased variant lists. HaplotagLR is user-configurable and includes a basic error model to control the empirical FDR in its output. We show that HaplotagLR outperforms the leading haplotagging method in simulated datasets, especially at high levels of specificity, and displays 7% greater sensitivity in haplotagging real data. HaplotagLR advances both the immediate utility of haplotagging and paves the way for further improvements to this important method.

## Introduction

Compared to the human reference genome, individual genomes differ at $\sim 4$–5 million sites [1]. These variants are known to affect many genomic processes [2] and their specific effects often depend on interactions with neighboring variants. Therefore, accurately predicting a variant's effects requires knowledge of which other variants are present in the host genome, particularly those on the same chromosomal homolog (in-cis). These relationships are captured by haplotypes, which describe the set of alleles present on a single chromosomal homolog. In diploid organisms, two haplotypes exist for each chromosome: one each for the maternal and paternal homolog, which typically differ at numerous individual variant positions. While modern methods make genotyping an individual across all known variant loci a reasonably trivial

**Funding:** This work was funded partially by a National Science Foundation CAREER Award (DBI-1651614) and NIH grants R21HG011493 and R01GM144484. C.C. was funded by the University of Michigan Rackham Merit Fellowship and the Training Program in Bioinformatics (T32GM070449). The funders had no role in study design, data collection and analysis, decision to publish, or preparation of the manuscript.

**Competing interests:** The authors have declared that no competing interests exist.

task, we cannot automatically discern the parental genome from which each allele originated. By extension, neither can we tell which alleles are in-cis on each parental homolog. To do so, we must apply a process known as phasing, which predicts maternal and paternal haplotypes using one of several statistically-based phasing methods [3–8]. Most contemporary methods employ read-based phasing, in which haplotypes are learned directly from sequencing reads. While read-based phasing utilities will accept reads from any sequencing protocol, phasing based on long-reads is significantly more reliable than short-read phasing [7, 9–12].

Read-based phasing tools typically provide phased variants in VCF format as output. These sets of phased variants are useful for many research applications, but are most-commonly associated with variant discovery, clinical genetics, and precision medicine analyses. For example, a recent publication describes the use of phased variants to interrogate structural variation at a single-cell level in cancer cells [13]. However, in recent years, phased variants are being increasingly used as markers in functional genomics to gain insight into relationships between genetic variation and observable phenotypes. Specifically, the haplotypic relationship between SNV alleles in an individual's genome can be used to identify biased contributions between parental alleles to sequencing reads in functional genomics assays. For example, genome-wide detection of allele-specific binding (ASB) at transcription factor (TF) binding sites (TFBS). In ASB, the parental allele(s) at variant position(s) within a TFBS lead to differential TF binding affinity between parental homologs [14]. It has been demonstrated that ASB contributes directly to observable variations in target gene expression [15] and significant enrichments of ASB loci are found among GWAS and disease-related SNPs [14], thus ASB has potential to shed light on any phenotype or disorder with a genetic-regulatory component. ASB is typically detected as a statistically-significant imbalance between reads originating from the parental homologs at a ChIP-seq binding locus. Haplotagging is a necessary first step in performing such analyses.

The haplotagging process is essentially the reverse operation of read-based phasing. Haplotagging labels each sequencing read with its parental homolog of origin based on the variants with which it intersects. Typically, given a phased VCF, a set of sequencing reads, and a statistical model, each sequence read is processed by: 1) gathering data for all variables in the model, usually including counts of matches and mismatches to two or more parental haplotypes, local and/or global sequencing error rates, etc., within the read's boundaries; 2) scoring of individual reads/dataset partitioning with the statistical model to predict the parental homolog from which it originated; and (optionally) 3) applying a scoring threshold and/or error model to control the empirical false-discovery rate. The result is reads that are labeled with their corresponding haplotype-of-origin and, ideally, a confidence score and accompanying metadata that can be used to evaluate the haplotagging decision retrospectively.

While haplotagging is gaining in popularity, we are aware of only two published haplotagging utilities: WhatsHap haplotag [7] and MarginPhase [16]. Both methods are maintained by the same group and are based on the same hidden-Markov model described in [16], which attempts to optimally partition reads between all parental haplotypes, labeling each read accordingly. According to the authors, the only differences between the two methods are in preprocessing steps, with MarginPhase being optimized for PacBio data and WhatsHap for Nanopore data [16]. Unfortunately, both are parts of larger software suites designed for read-based phasing; MarginPhase in particular is only capable of haplotagging within the context of end-to-end read-based phasing. Critically, both appear to be targeted at the traditional variant discovery and clinical genomics audiences. A common feature of most clinical genomics studies is a focus on newly-discovered variants in patient samples or primary cell lines. In these cases, the variants themselves are the focus, and novel variants are typically used to improve existing phased haplotypes through read-based phasing. Basic research studies, by contrast, are

typically carried out within well-described cell lines for which high-confidence variant sets are readily available. In these cases, phased variants are used primarily as a marker of a read's parental origin, while the variants themselves may not be a direct focus. Thus, full read-based haplotype construction and phasing in all these studies would be computationally wasteful in much the same way it would be wasteful to require all researchers to construct reference genomes from their own read data.

Furthermore, both WhatsHap haplotag and MarginPhase present shortcomings from both technical and usability standpoints. Foremost among these, we could not find published information on how to interpret the numerical scores reported in their output, making it difficult to evaluate the quality of each prediction. Neither are estimates of false-discovery and/or error rates provided and users are given no control over model parameters and/or scoring thresholds within the context of the haplotagging utility itself. No data are given for variables used in haplotagging decisions, thus results give little insight regarding the underlying sources of haplotagging error. Finally, little guidance is given in the published literature on how to apply these tools to perform haplotagging as a standalone operation and, in particular, how to interpret haplotagging confidence and control the false discovery rate within results. Despite their popularity, these shortcomings represent significant obstacles to broad use of these tools as parts of basic research genomics pipelines. Therefore, a user-friendly haplotagging method based on a statistically-rigorous, tunable model, offering control over error rates, and providing sufficient metadata in its output to retrospectively analyze results for clues into the causes of haplotagging error, would represent a significant advancement.

We present HaplotagLR: a haplotagging utility designed specifically for long sequencing reads. HaplotagLR uses a multinomial model that integrates phased variant data and sequencing error effects to haplotag reads. HaplotagLR was designed to be a user-friendly and configurable alternative to current methods, offering investigators control over the tradeoff between sensitivity and specificity through inclusion of tunable model parameters and scoring thresholds, to give control over false-discovery and error rates. We show that HaplotagLR outperforms WhatsHap haplotag, particularly in sensitivity, exhibiting greater recall at comparable precision in simulated data analyses and haplotagging a larger proportion of biological nanopore sequencing reads, and that the optional negative-binomial error model is able to meaningfully and predictably reduce the prevalence of haplotagging errors in its output. Finally, since its output includes all data used in haplotagging decisions, it can be used to directly analyze the underlying causes of haplotagging error, something that remains largely unexplored in the published literature.

## Materials and methods

### Description

HaplotagLR is a command-line utility for haplotagging long sequencing-reads based on phased heterozygous variants from all contributing genomes, for example, the maternal and paternal genomes of a diploid organism (Fig 1). HaplotagLR requires three inputs: long sequencing reads, which may be mapped (BAM format) or unmapped (fastq format), a reference genome (fasta format), and fully-phased variant data (VCF format). HaplotagLR is specifically designed to process long sequencing reads, and is applicable to reads from any common long-read sequencing platform, including Nanopore and PacBio. Genomic DNA fragments may be isolated from any cells with available phased variants, in VCF format. Mapped reads may be prepared in advance using any mapping and filtering pipeline prior to processing with HaplotagLR. Alternately, HaplotagLR is capable of handling inputs in fastq format by applying an initial mapping step with Minmap2 [11], with default parameters, (Fig 1A) prior to

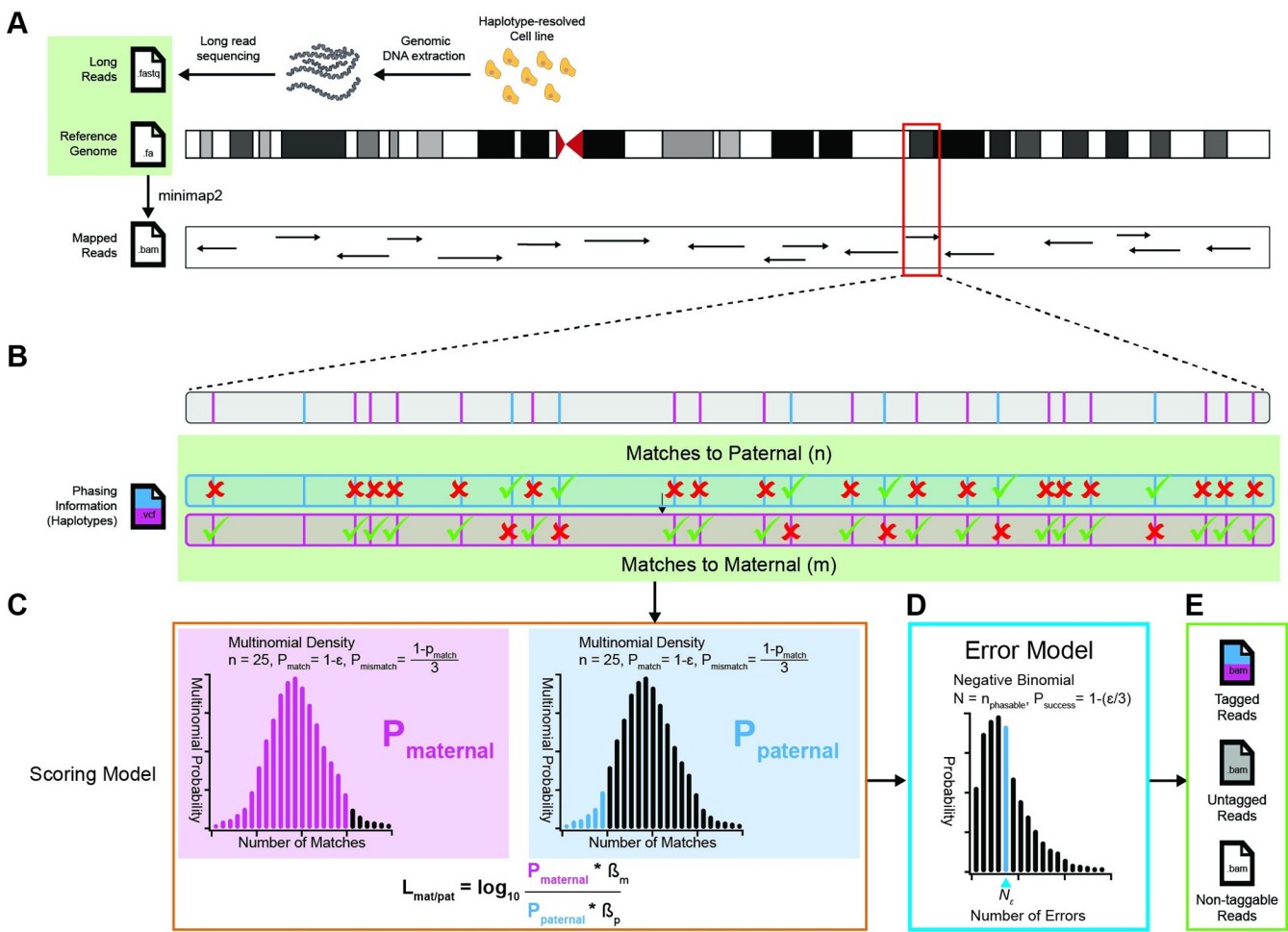

**Fig 1. Overview of HaplotagLR.** A. Long sequencing reads are first prepared from genomic DNA isolated from cells with available phased variant data. Reads may be mapped externally or processed with HaplotagLR, through an internal call to minimap2. B. BAM reads are intersected with phased, single-nucleotide variants (SNVs) supplied in standard VCF format. Match and mismatch counts are compiled for maternal and paternal phases and plugged into the scoring model. C. Match and mismatch counts are used to calculate multinomial probabilities for matches to maternal and paternal phases ($P_{maternal}$ and $P_{paternal}$ respectively). D. After all reads are haplotagged, the false-discovery rate (FDR) may be controlled by applying the optional haplotagging error model. E. Finally, tagged, untagged, and nontaggable reads are written to one or more output files in standard BAM format.

haplotagging, whereas pre-mapped reads in BAM format are processed directly with the haplotagging algorithm (Fig 1B–1D).

## The haplotagging algorithm

The HaplotagLR algorithm is described in Fig 1B–1D. Haplotagging begins by intersecting individual mapped reads with known phased, heterozygous single-nucleotide variants (SNVs), supplied as a standard VCF file. The number of matches and mismatches are counted for all phases (Fig 1B) and these are then plugged into the scoring model (Fig 1C), which is discussed in the "Scoring Model" section, to calculate log-likelihood ratios (LLRs) for each haplotype. After scoring, the haplotype with the highest LLR is identified and the read is haplotagged accordingly. Reads overlapping no heterozygous variants are labeled as "untaggable," since they contain no information by which to match to any haplotype, while those where ties are present between haplotypes are labeled "untagged". Optionally, the haplotagging error model (Fig 1D), described in the "Haplotagging error model" section, may be applied to enforce a

specific FDR. Finally, tagged, untagged, and untaggable reads are written to one or more output files, with BAM tags added to capture all available variables used in calculating LLRs and choosing the best haplotag (Fig 1E). These tags are described in detail within the HaplotagLR github documentation.

## Scoring model

The goal of HaplotagLR is to assign all input reads to one of the known haplotypes specified in the input vcf file. Each read is assumed to have originated from a physical DNA source and the name of this genomic source (i.e.: the name of the cell line that the DNA submitted to sequencing was extracted from) is defined as the sample.

Model notation is summarized in Table 1. A vcf file can contain variants from multiple samples, and we define S as the set of all samples in the input vcf file. For a given sample $s$ in a vcf file, there may be variants belonging to multiple phase sets, where a phase set is a set of phased variants that are phased relative to one another. Let $P_s$ be the set of all phase sets in sample $s$ for all $s \in S$. For each phase set $p \in P_s$, let $A_p$ be the set of all phased variants specified by the vcf. The set of haplotypes for phase set $p$ is defined as $H_p = \{h_1, \ldots, h_l\}$, where $l$ is the ploidy of the phase set (ie: number of distinct sequences existing in the sample population at that genomic position). For all $h_i \in H_p$, let $h_i[a]$ denote the allele for haplotype $i$ at phased variant $a$ for all $a \in A_p$.

Input reads must be labeled with the sample from which they originated (see methods section). Let R be the set of all input reads, $R_{mapped}$ be the set of all mapped reads in R, and for all $s \in S$ let $R_s$ be the set of all reads that originate from sample $s$ in $R_{mapped}$, and let $R_{s,phasable}$ be the set of all reads in $R_s$ that overlap at least one phased variant. For each read $r \in R$, let $B_r$ be the set of all called bases for that read, let $E_r$ be the set of sequencing error rates for each called base in read $r$. For all $r \in R_{s,phasable}$, let $\underline{b}_r[a]$ denote the aligned read base, and let $e_r[a]$ denote the sequencing error rate for the aligned read base at phased variant $a$ for all $a \in A_p$.

LRphase scores follow a multinomial distribution whereby the chance of observing a match between $b_r[a]$ and $h_i[a]$ are proportional to the average sequencing error rate ($\varepsilon$) in the read data; by default, HaplotagLR obtains the estimated sequencing error rate directly from each BAM record's 'de' tag. HaplotagLR also provides the ability to use a user-specified global value for epsilon, which is then used in both the scoring and error models.

We posit that, given that we have observed a match between $b_r[a]$ and $h_i[a]$, the probability of a true match between the original sequence and $h_i[a]$ is equal to (1-$\varepsilon$). Likewise, given an observed mismatch between $b_r[a]$ and $h_i[a]$, the probability of a true mismatch between the

**Table 1. HaplotagLR model definitions.**

| Set | Total | | Vector Info |
|---|---|---|---|
| $S = \{s_1, \ldots, s_c\}$ | $c$ = total samples in vcf | | |
| $P_s = \{p_1, \ldots, p_d\}$ | $d$ = total phase sets in a sample s | | |
| $A_p = \{a_1, \ldots, a_n\}$ | $n$ = total phased variants in phase set $p$ | | |
| $H_p = \{h_1, \ldots, h_l\}$ | $l$ = ploidy of haplotype for phase set $p$ | $h_i[a]$ | Allele for haplotype $i$ at phased variant $a$ |
| $R_{s,phasable} = \{r_1, \ldots, r_j\}$ | $j$ = total mapped reads for sample $s$ with at least one phased variant overlapped | | |
| $B_r = \{b_1, \ldots, b_k\}$ | $k$ = total number of called bases for read $r$ (i.e., the read length) | $b_r[a]$ | Read base observed at phased variant $a$ |
| $E_r = \{e_1, \ldots, e_k\}$ | $k$ = total number of called bases for read $r$ (i.e., the read length) | $e_r[a]$ | Sequencing error rate for read base at phased variant $a$ |

original sequence and $h_i[a]$ is equal to $\left(\frac{\varepsilon}{3}\right)$ since there are three possible incorrect nucleotides at position $a$. Therefore, the overall probability that a read containing $n$ heterozygous variants originated from haplotype $h_i$ is given by Eq 1.

Overall probability of a match between read $r$ and haplotype $i$

$$P(h_i) = \left(\frac{n!}{k_{n-m}!k_m!}\right) \times \left(\frac{\varepsilon}{3}\right)^{k_{n-m}} \times (1-\varepsilon)^{k_m} \tag{1}$$

where $k_m$ is the number of variants matching $h_i$, and $k_{n-m}$ is the number of variants mismatching $h_i$ within the aligned read, and the first model term is the standard multinomial coefficient, which may be disabled via a command-line option if desired.

In order to accommodate variation in the sequencing error rate between base positions, we generalize this model by applying the per-base sequencing error rate, $e_r[a]$ for the global sequencing error rate, $\varepsilon$, at each heterozygous variant position, as described in Eq 2. For convenience and efficiency, this calculation is done in log-space.

Generalized overall probability of a match between read $r$ and haplotype $i$

$$log10(P(h_i)) = \sum_{l=1}^{n} \left[ log10(1 - e_r[a])x_a + log10\left(\frac{e_r[a]}{3}\right)y_a \right] + log10\left(\frac{n!}{k_{n-m}!k_m!}\right) \tag{2}$$

Wherein $n$ represents the total number of heterozygous positions within read $r$, $m$ represents the number of observed matches to haplotype $i$, and $x_a$ and $y_a$ are indicator variables that take the value 1 when in the presence of a match ($x_a$) or mismatch ($y_a$) between the observed nucleotide for variant $a$ in read $r$ to the corresponding variant record in haplotype $i$, and 0 otherwise. Finally, $e_r[a]$ represents the per-base error probability, which is calculated from the BAM phred-scaled quality scores for each read according to Eq 3, wherein $q_l$ represents the per-base phred score, thus capturing local variations in sequence quality.

Per-base sequencing error rate calculation

$$e_r[a] = 10^{-0.1 \times q_l} \tag{3}$$

Multinomial probabilities are then adjusted by uniform Bayesian prior probabilities for each haplotype $i$ (Eq 4):

Uniform Bayesian prior calculation

$$\beta(h_i) = \frac{1}{l} \tag{4}$$

For all $h_i \in H_P$. This implies an equal probability of observing any given haplotype a-priori. The posterior probability that the observed sequence of variants matches $h_i$, then, follows Eq 5:

Posterior probability of a match between read $r$ and haplotype $i$

$$P(r)_{h_i} = \frac{P(h_i)\beta(h_i)}{\left[\sum_{j=1}^{l} P\left(h_j\right)\beta\left(h_j\right)\right] - P(h_i)\beta(h_i)} \tag{5}$$

Phasing decisions are based on the likelihood ratio of haplotype $h_i$ vs all other haplotypes in the model (Eq 6). When phasing reads, LRphase calculates $LR_{h_i}$ for all haplotypes $h_i \in H_P$. Reads are assigned to the phase corresponding to $max(LR_{h_1}, \ldots, LR_{h_l})$ and this value is used as the observed score for the read.

Likelihood ratio calculation

$$LR_{h_i} = \frac{P(r)_{h_i}}{\left[\sum_{j=1}^{l} P(r)_{h_j}\right] - P(r)_{h_i}} \tag{6}$$

## Haplotagging error model

Since haplotagging is primarily based on the number of matches and mismatches to each phase, we expect errors in these counts to contribute to most incorrect haplotags. Many counting errors can be explained by corresponding sequencing errors in the long-read data, which occur at an average rate ε, which may vary locally. Sequencing errors render the observed nucleotide at position $l$ uninformative when the substituted base does not match either parental phase. Therefore, a haplotagging decision based on variant $l$ may be reduced to a random guess between maternal and paternal phases with probability proportional to ε. Assuming sequencing errors are randomly distributed among reads and occur at equal rates among heterozygous sites, and because there are three possible "incorrect" nucleotides at each position, each variant contributes ε/3 to the expected maximum phasing error rate: $(\varepsilon/3)^n$, where $n$ equals the number of variants within a sequence. In practice, we ignore the number of variants and simply use ε/3 as a conservative estimate of the phasing error rate.

Given these simplifying assumptions, we consider haplotagging decisions as Bernoulli trials with $P_{success} = 1-(\varepsilon/3)$ and model the error distribution as a negative binomial with $N$ = the number of phaseable reads. Thus, to control the error rate at any given FDR, we use the mean of this distribution, $N_\varepsilon$, as the expected number of phasing errors, and reassign the $N_\varepsilon^*$ $(1-FDR)$ lowest-scoring reads as "unphased." Since it is independent of the scoring model, this error model has the advantage of being applicable to both of HaplotagLR's scoring models. By default, the average sequencing error rate is calculated empirically by taking the mean of observed sequencing error rates across all sequencing reads processed.

## Data simulation

Simulated long reads were used to assess the performance of HaplotagLR compared to WhatsHap haplotag, the current leading haplotagging utility. The phased VCF for NA12878 (HG001) was retrieved from the Genome In A Bottle (GIAB) consortium repository [17] (S1 File).

The GIAB VCF file contained anomalies that elicited run-time errors from both HaplotagLR and WhatsHap haplotag. This VCF is noncompliant with the VCF 4.3 specification (https://samtools.github.io/hts-specs/VCFv4.3.pdf) in at least two ways: 1) Phase Set (PS) tags within genotype fields contain strings instead of 32-bit integers. 2) Sample columns in the VCF column header are labeled"INTEGRATION" rather than containing the sample name. We also found ∼25,000 records with malformed genotype records, which contained extra fields not defined in the format string. Finally, we encountered errors from WhatsHap that suggested at least a subset of indel records are also malformed. Neither program would run successfully without correcting these errors, but we were able to rescue the VCF using a custom Python script. Briefly, we transliterated PS tag strings to integer values by concatenating a unique integer with the chromosome number for each record (24, 25, and 26, for chromosomes X, Y, and M, respectively). This ensured that all phase sets have integer labels and only included variants on the same chromosome. We defined an additional format tag, OPS, under which we stored the original PS tag values. Likewise, the "INTEGRATION" label in the column name header was replaced with the sample name, "HG001". Variants with malformed

genotype fields were rescued by removing the fields not defined in the format string. Since we were unable to identify the direct cause for WhatsHap errors related to indel record parsing, we filtered out all records for indels and structural variants, leaving only SNV records in the VCF. The rescued and filtered VCF and scripts to produce it from the original are archived at https://zenodo.org/records/10257128 (S1 File).

A modified version of HaplotagLR was used to simulate reads for a single hypothetical human genome with maternal and paternal phasing information. Briefly, haplotype-specific reference sequences were generated with 'bcftools consensus'[18] based on the rescued GIAB VCF and hg38 human reference sequence. Each haplotype-specific fasta was fed separately into pbsim2 [19] as the reference from which simulated reads were randomly drawn, up to 1X coverage. Parameters controlling the read length distribution, sequencing, and base calling error rates were set to emulate typical performance of the MinIon sequencing platform with flow cell version R10.4.1 (https://nanoporetech.com/products/minion). These are as follows: '—depth 1 –hmm_model R103.model—difference-ratio '23:31:46'—length-mean 25000—length-min 100—length-max 1000000 –length-sd 20000—accuracy-mean 0.98—accuracy-min 0.01—accuracy-max 1.00'. Simulated reads were aligned to the hg38 reference genome with minimap2 [11] and correct haplotags and alignment coordinates were encoded in the read names. Finally, samtools [20] was used to remove duplicated and supplementary reads, and concatenate, sort, and index reads into a single combined bam file. Of 258,539 total reads, 246,210 were mappable, and 178,504 overlapped at least one heterozygous variant in HG001. The modified version of HaplotagLR and bam alignment of simulated reads are available at https://zenodo.org/record/7823582. Public URLs for all source data and simulated datasets produced in this study are summarized in S1 File.

## Haplotagging simulated Oxford nanopore sequencing data

Simulated reads were haplotagged using HaplotagLR haplotag mode with the following arguments: '/usr/bin.time HaplotagLR haplotag -i -COMBINED_alignment.dedup.nosup.sorted.bam v phased_variants.vcf.gz -r hg38.fa -O combined -F 0—log_likelihood_threshold 0 -t 1 -o. -s HG001'. For comparison, WhatsHap [7] was used in 'haplotag' mode to phase simulated reads with the following arguments: '/usr/bin/time whatshap haplotag—ignore-read-groups—skip-missing-contigs—reference hg38.fa phased_variants.vcf.gz COMBINED_alignment. dedup.nosup.sorted.bam -o HG001.COMBINED.whatshap_phasing_results.bam'. This run did not employ the optional Phasing Error Model, therefore no scoring threshold was applied and all phaseable reads are reported in the output. Because MarginPhase and WhatsHap share the same scoring model, differing only in preprocessing steps, we did not include it in the comparison of methods. The /usr /bin/time utility was used to gather statistics on runtime and peak memory usage for both programs but runtimes were not directly comparable since HaplotagLR is implemented in Python, whereas WhatsHap is implemented in C++.

Haplotagged results from both utilities were compared to the base truth by comparing 'HP' tag values for each BAM record to the correct haplotag, which was encoded as part of each read's name string. Correct haplotag assignments were counted as true-positive predictions. False positive predictions were defined as incorrectly haplotagged reads intersecting at least one heterozygous variant (phaseable reads). Reads without any overlapping heterozygous variants (nonphasable reads) and assigned the "nonphasable" label were counted as true negatives. Nonphasable reads assigned to either haplotype would constitute false positive predictions. However, these are not observed in practice, since neither algorithm will phase reads without intersecting heterozygous variants. Phaseable reads labeled "unphased" were counted as false-negative predictions. Rates of true and false positives and negatives were tabulated for both

methods and used to calculate Precision-Recall (PR) curves (Fig 2) and PR-AUC was used as an overall performance measure.

Additional runs with the optional Haplotagging Error Model were performed to obtain basic performance metrics using two different FDR thresholds: 0.1 (the default) and 0.2), using commands 'HaplotagLR phasing -i -COMBINED_alignment.dedup.nosup.sorted.bam v phased_variants.vcf.gz -r hg38.fa -O combined_fdr10 -F 0.1 -t 1 -o. -s HG001', and 'HaplotagLR phasing -i -COMBINED_alignment.dedup.nosup.sorted.bam v phased_variants.vcf.gz -r hg38.fa -O combined_fdr20 -F 0.2 -t 1 -o. -s HG001'. Since WhatsHap haplotag lacks any configuration options we were unable to optimize scoring/error thresholds for comparison.

## Haplotagging biological Oxford nanopore sequencing data

To assess the performance of HaplotagLR on biological sequencing data, we used a previously published nanopore sequencing dataset from our lab [21]. This dataset contains 2.0GB of sequence in 123.3k spots, available from SRA under the accession number SRR13615770 (S1 File). Basecalled data in fastq format were mapped to the hg38 reference genome using minimap2 and output bam alignments were indexed with samtools. The resulting bam file contained 7,088,481 total, 7,016,931 mapped, and 307,281 supplementary reads. Mapped and indexed reads were analyzed with HaplotagLR phasing mode, without the error model, and WhatsHap haplotag mode. HaplotagLR was run with the following arguments: 'HaplotagLR phasing -v phased_variants.vcf.gz -i FAS75579.Nanopore.sorted.bam -r hg38.fa -O combined -F 0—log_likelihood_threshold 0 -t 1 -o. -s HG001'. WhatsHap was run with the following arguments: 'whatshap haplotag—ignore-read-groups—no-reference phased_variants.vcf.gz FAS75579.Nanopore.sorted.bam—skip-missing-contigs -o HG001.COMBINED.whatshap_phasing_results.bam—output-haplotag-list HG001.COMBINED.whatshap_phasing_results.txt'.

## Results

### Comparison to WhatsHap haplotag

HaplotagLR was evaluated in comparison to the 'haplotag' mode of WhatsHap [7]. Performance was compared using a set of simulated long reads consisting of 178,504 reads overlapping at least one heterozygous, phased variant in the HG001 genome. Both methods exhibited good performance with very low empirical rates of haplotagging error but HaplotagLR demonstrated superior recall, correctly haplotagging 99.5% of phaseable reads (i.e., those overlapping at least one heterozygous variant) compared to 98.1% for WhatsHap. To facilitate direct performance comparisons, we defined false-positives as incorrectly haplotagged phaseable reads and false-negatives as phaseable reads not assigned to either haplotag. The greater sensitivity we observed for HaplotagLR was also reflected in the empirical rates of incorrect and false-negative haplotagging assignments (IHR and FNR, respectively): HaplotagLR IHR = 0.48%, FNR = 0.42%; WhatsHap IHR = 0.27%, FNR = 1.7%. Overall, HaplotagLR outperformed WhatsHap by a slight margin when comparing precision-recall curves (HaplotagLR PR-AUC = 0.999993; WhatsHap PR-AUC = 0.999954), retaining nearly 100% precision up to ∼95% recall (Fig 2A) compared to ∼90% recall for WhatsHap (Fig 2B).

For both methods, runtime was a bigger concern than memory usage, with both methods showing a negligibly small memory footprint when applied to the simulated dataset. However, HaplotagLR memory usage (138,336 kb) was had substantially lower than WhatsHap haplotag (801,684 kb). In terms of runtime, HaplotagLR required 10,216.7 seconds of user time to haplotag all reads compared to 440.2 seconds for WhatsHap haplotag, an approximately 23-fold handicap. However, this comparison is not particularly meaningful given that HaplotagLR is implemented in Python whereas WhatsHap is implemented in C++.

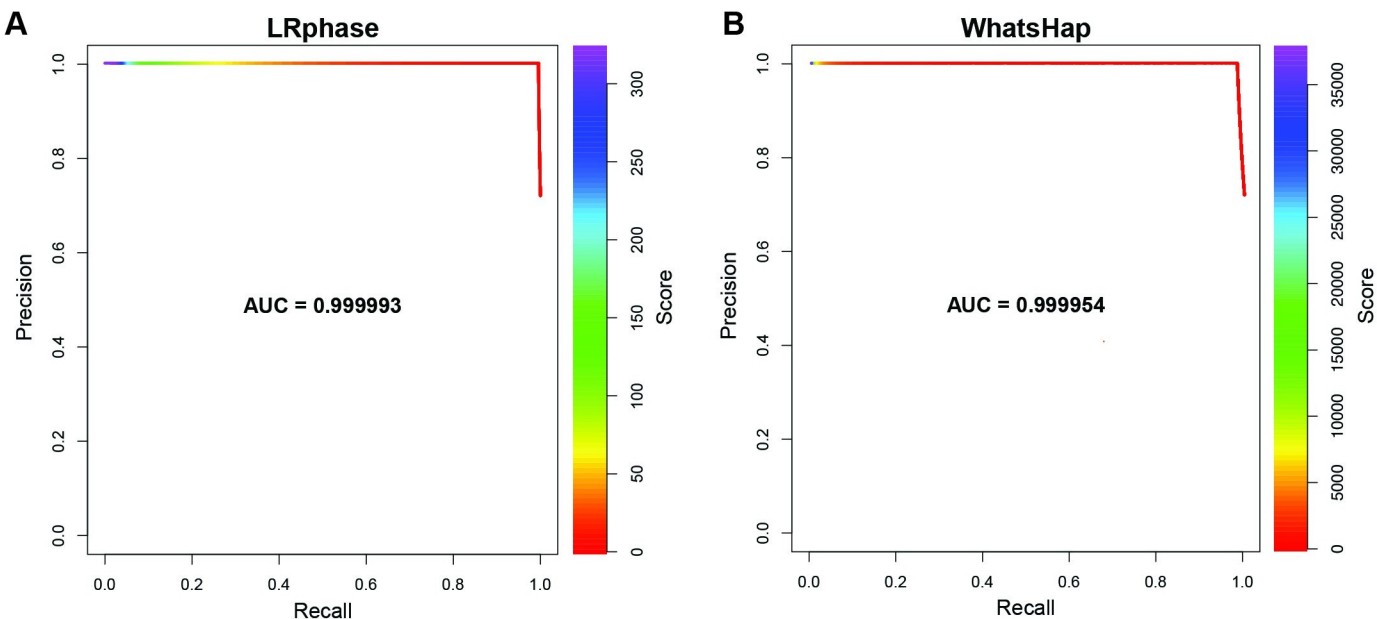

**Fig 2. Haplotagging performance of HaplotagLR and WhatsHap.** Precision-Recall (PR) curves are used to assess the ability of HaplotagLR (A) and WhatsHap (B) to identify the correct haplotag. Curves describe the fraction of correct and incorrect haplotag assignments across varying score thresholds. Individual points along each curve are color coded to illustrate score ranges at each combination of precision and recall. A. PR curve for HaplotagLR haplotag assignments. B. PR curve for WhatsHap haplotag assignments.

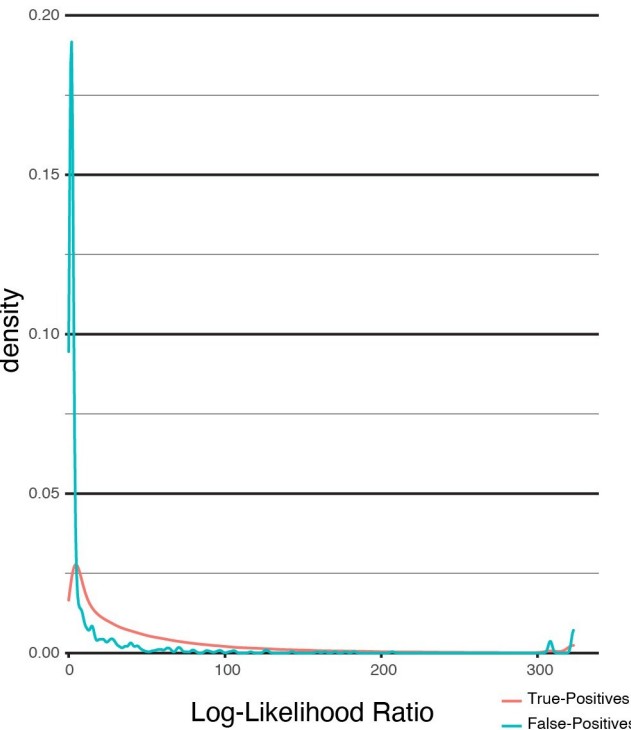

**Fig 3. Rates of correct and incorrect haplotagging assignments relative to log-likelihood ratio score.** The X axis represents the observed log-likelihood ratio used by HaplotagLR to assign reads to a phase. The Y axis represents the density of True-Positive and False-Positive observations at any given score. Traces show the smoothed gaussian kernel density estimates for true-positive and false-positive read assignments.

## Evaluation of the error model

We noted in our analysis of simulated data that the overwhelming majority of haplotagging errors occur at log-likelihood ratios between 0 and 1 (Fig 3). Indeed, the occurrence of haplotagging errors showed a significant inverse-correlation with LLR score ranking (Kruskal-Wallis H-test p = $1.3 \times 10^{-215}$). Accordingly, truncating the phasing assignments below a small LLR value should preferentially remove false-positive haplotags. Indeed, when HaplotagLR is run with the error model at the default 10% FDR cutoff, we observed the expected decrease in IHR at the expense of increased FNR (HaplotagLR IHR = 0.35%, FNR = 1.8%). While this degree of increase in FNR is substantial in comparison to HaplotagLR without the error model, the overall effect was to render overall performance roughly comparable to WhatsHap haplotag. Relaxing the FDR threshold to a more-permissive 20% yielded a marginal decrease in FNR of 0.5 percentage points while IHR showed a marginal increase of only 0.03 percentage points (HaplotagLR IHR = 0.38%, FNR = 1.3%). This was consistent with our expectations and confirmed that tuning the FDR parameter meaningfully and predictably impacts the balance between IHR and FNR in haplotagged results.

## Performance on biological nanopore sequencing reads

Unfortunately, no experimentally-validated gold-standard set of haplotagged long sequencing reads was available to directly assess the performance and accuracy of HaplotagLR or WhatsHap haplotag when applied to biological sequencing data. Therefore, we applied HaplotagLR and WhatsHap haplotag to a biological nanopore sequencing dataset, generated in-house [21], consisting of 179,398 long reads, of which 169,609 were mappable to the hg38 genome, and 75,702 overlapping at least one heterozygous variant in HG001. Since correct haplotags for reads within this dataset were unknown prior to haplotagging, we were only able to compare the fraction of reads haplotagged between the two methods. In total, HaplotagLR haplotagged ~97.6% of taggable reads compared to ~91.2% for WhatsHap. This increased sensitivity relative to WhatsHap may be an advantage in applications where it is critical to maximize read retention.

## Discussion

Haplotagging tools were originally designed by researchers primarily focused on directly studying novel variation, often with the goal of identifying variants correlated with a specific phenotype or genetic disorder of interest. In these assays, variants are usually discovered de-novo within primary samples where variation is not well-characterized. Such analytical workflows begin with a variant discovery step, after which novel and known variants are combined to improve existing phased haplotypes through full-fledged read-based phasing. These improved haplotypes may or may not be used to individually haplotag the observed sequencing reads depending on the goals of the analysis and this step is often skipped in studies focused on haplotypes rather than individual variants. It is easy to understand how the value of haplotagging as part of functional genomics analyses has gone largely unrecognized.

More recently, haplotagging has come to the attention of functional and evolutionary genomics researchers, largely because of its power to show correlations between individual variation and specific types of functional variability, as in the allele-specific binding example. In these cases, variations within sequencing reads are used primarily as a marker of parental origin and the processing pipeline typically include neither variant discovery nor read-based phasing steps. Rather, known variants, pre-processed into fully-phased haplotypes, are normally used to haplotag reads. In these cases, packages like WhatsHap and MarginPhase are

overly complex; this inherent complexity presents a barrier to their proper use and invites redundant analysis.

Because a variants' functionality often depends on interactions with other variants on the same chromosome, accurately predicting these effects requires knowledge of which alleles are present at other variants in-cis. For example, loss-of-function through compound heterozygosity can only occur when two non-reference SNV alleles exist on separate homologs of the same functional element, leaving no functional copy capable of rescuing the mutant phenotype. By contrast, cis-regulatory variants are typically restricted to interactions with those variants with which they are in-cis. Therefore, haplotagging is necessary to predict how the regulatory variants carried by an individual will interact with each other, and with variants carried by their target genes. Labeling reads with their haplotype of origin by haplotagging makes it trivial to identify biases in sequencing data caused by such effects, thus facilitating follow-up analyses to gain clues regarding global mechanisms governing gene expression, natural selection, and the etiology of genetically-based disorders. Therefore, continued development of reliable and user-friendly haplotagging tools should be an important goal in genomics.

HaplotagLR has better overall performance than WhatsHap haplotag. Our results show that using HaplotagLR with default error model settings (10% FDR) yields performance metrics comparable to WhatsHap haplotag when applied to the same data. We also show that HaplotagLR, when run without the error model or with a relaxed FDR threshold, outperforms WhatsHap haplotag, which offers no ability to tune model parameters or scoring thresholds and presents only a numerical score without context. Importantly, HaplotagLR gives users control over haplotagging decisions through multiple options allowing configuration of parameters and score thresholds for both the scoring and error models. Furthermore, all data directly used to calculate LLRs and make haplotagging assignments are included in the output for use in downstream analyses. For example, retrospective analysis of haplotagging decisions may allow us to comprehensively describe the causes of haplotagging error, an area that is poorly explored in the published literature. This would greatly advance our ability to construct improved scoring and error models, with the goal of reliably separating true-positives from false-positives based only on their ranked scores.

Indeed, the biggest limitation of HaplotagLR is the inability of the scoring model to reliably separate true-positives from false-positives, particularly at low LLR values. These often occur when few heterozygous sites are found within a read, in stretches of low-quality sequence, or in poorly-aligned regions. While we have shown that phasing errors are, as expected, concentrated in the lower extreme of the scoring distribution, the lowest-scoring reads, as a group, appear to be a mixture of true-positives and false-positives that currently do not reliably stratify by score. The consequence is that, presently, reductions in FDR will always be accompanied by a loss of sensitivity. Interestingly, despite using very different underlying models and algorithms to assign haplotags, WhatsHap haplotag and HaplotagLR with the error model performed almost identically, suggesting that both models are subject to the same sources of error and neither method adequately captures these sources in its model at present.

We hypothesize that, in addition to sequencing error, haplotagging performance will also be affected by errors in variant discovery, genotyping, haplotype construction, and variant phasing, as well as errors introduced in alignment and post-processing of sequenced reads. However, this list may not be exhaustive and our knowledge of precisely how these factors contribute to haplotagging error is sparse. Incorporating these factors into improved models will require exploration of the relationships between them and incorrect haplotag assignment. Thus neither strategy, at present, appears to offer a distinct advantage in its ability to reduce FDR without sacrificing sensitivity when based only on score rankings. It is our hope that including additional metadata from both the input BAM and VCF files in haplotagLR output

will facilitate comprehensive review of how these factors relate to haplotyping errors. Meanwhile, we include the negative-binomial error model as an option in the present implementation to offer users some control, albeit imperfect, over the prevalence of haplotagging errors in their results. Since false-positive haplotags are concentrated at low LLRs, the majority can be excluded by setting an aggressive scoring threshold in experiments whenever accurate haplotags are critical. Conversely, setting the FDR and/or log-likelihood thresholds to zero will force every read with sufficient heterozygous site data to be haplotagged; this is applicable when read retention must be prioritized. Another limitation of the current version is that it is capable of handling only diploid genomes. However, as discussed in the methods section under "Scoring model", the model itself readily generalizes to a polyploid case and we have plans to implement this functionality in a future release.

Our results suggest that the scoring models used by HaplotagLR, WhatsHap haplotag, and MarginPhase, could all be greatly improved given a better understanding of the causes of haplotagging error, something that remains understudied in the literature. While we know that variant calling and/or phasing quality scores, when present in the VCF input file, may be informative, getting at the direct causes of haplotagging error will require systematic retrospective analysis of haplotagged datasets. This is only possible with comprehensive knowledge of the variables examined in the haplotagging decision. Importantly, HaplotagLR output provides much of this information already, and includes sufficient detail to extract all remaining variables used in scoring and tagging each read from the input files. Specific tags added to each BAM record are documented within the github documentation for HaplotagLR. Insights derived from retrospective analyses may then be used to improve scoring and/or error models capable of stratifying true-positive and false-positive predictions based on their scores alone. Ideally, this would involve analysis of both simulated and biological datasets. In particular, a gold-standard set of experimentally-validated haplotagged reads would be of great use in developing reliable haplotagging methods. HaplotagLR, given its comprehensive output, offers the ability to interrogate haplotagged results to systematically identify the causes of haplotagging errors.

HaplotagLR has potential applications in diagnostics, clinical, and basic research into disease risk, gene regulation, and population-level genomic variation. These include targeted sequencing, phasing cut sites, high-throughput diagnostics, experimental design and hypothesis formulation, and investigation of ASB and/or DNA modification (methylation, etc.). Given its flexibility and ease of use, HaplotagLR can be dropped into most analytical pipelines, opening haplotagging to a broad audience.

## Supporting information

**S1 File. Data locations.**
(XLSX)

## Acknowledgments

We would like to thank Carolyn Boyle and the University of Michigan Consulting for Statistics, Computing and Analytics Research (CSCAR) for providing statistical consultation in development of the scoring and error models, and members of the Boyle lab for editorial support and suggestions on the manuscript.

## Author Contributions

**Conceptualization:** Monica J. Holmes, Babak Mahjour, Christopher P. Castro, Alan P. Boyle.

**Formal analysis:** Monica J. Holmes, Babak Mahjour, Christopher P. Castro, Gregory A. Farnum, Adam G. Diehl.

**Funding acquisition:** Alan P. Boyle.

**Investigation:** Adam G. Diehl.

**Methodology:** Monica J. Holmes, Babak Mahjour, Christopher P. Castro, Gregory A. Farnum, Adam G. Diehl, Alan P. Boyle.

**Software:** Monica J. Holmes, Babak Mahjour, Christopher P. Castro, Gregory A. Farnum, Adam G. Diehl.

**Supervision:** Alan P. Boyle.

**Validation:** Adam G. Diehl.

**Writing – original draft:** Adam G. Diehl, Alan P. Boyle.

**Writing – review & editing:** Adam G. Diehl, Alan P. Boyle.

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
