## [Decision Letter · Decision Letter 0]

1 Oct 2023

PONE-D-23-26110LRPhase: an efficient and configurable utility for haplotagging long readsPLOS ONE

Dear Dr. Boyle,

Thank you for submitting your manuscript to PLOS ONE. After careful consideration, we feel that it has merit but does not fully meet PLOS ONE’s publication criteria as it currently stands. Therefore, we invite you to submit a revised version of the manuscript that addresses the points raised during the review process.

We look forward to receiving your revised manuscript.

Kind regards,

Pietro Cinaglia

Academic Editor

PLOS ONE

Journal Requirements:

3. Please expand the acronym “NIH” (as indicated in your financial disclosure) so that it states the name of your funders in full.

Reviewers' comments:

Reviewer's Responses to Questions

**Comments to the Author**

1. Is the manuscript technically sound, and do the data support the conclusions?

Reviewer #1: Partly

Reviewer #2: Yes

2. Has the statistical analysis been performed appropriately and rigorously? 

Reviewer #1: Yes

Reviewer #2: No

3. Have the authors made all data underlying the findings in their manuscript fully available?

Reviewer #1: No

Reviewer #2: Yes

4. Is the manuscript presented in an intelligible fashion and written in standard English?

Reviewer #1: Yes

Reviewer #2: Yes

5. Review Comments to the Author

Reviewer #1: The authors present a conceptionally simple statistical approach and a corresponding tool LRPhase to assign long reads to haplotypes defined by a given set of phased variants. This process is called haplotagging. LRPhase allows to specify an FDR and thus to adjust specificity and sensitivity according to application.

The manuscript is well written and the aim of haplotagging and potential application areas are well described. LRphase is applied on simulated data from Genome in a Bottle (GIAB) and on real data (ONT sequencing data). Comparison was performed with the haplotag mode of whatshap using precision-recall (PR) curves and PR-AUC on one simulated data set and percentages of haplotagged reads for one real data set.

The authors made an effort to provide data in reproducible and convenient ways using SRA, Github, Zenodo and Bioconda. All data are available except helper scripts for file fixing and data simulation. Although the corresponding analysis steps are explained in detail in the manuscript, I strongly encourage providing these scripts as well.

I think the described model and tool is useful, since haplotagging, i.e. read phasing according to a given set of phased variants, will gain importance as long-read sequencing is used not only for variant phasing and assembly, but increasingly also for the generation of downstream information such as targeted deep sequencing or RNA sequencing. Particularly, improving models to consider base calling error rates, as done here, as well as reporting characteristic numbers reflecting the specific result and quality of individual phased reads can be helpful in some application settings. However, there are some possible major shortcomings that need to be addressed. Further, there are missing results and information that will need to be added to put the work in relation to existing work and to make sure that the current version of the tool is indeed capable of addressing the issues discussed in the manuscript. I will start with a conceptional issue that I think needs to be addressed in the manuscript followed by further major concerns and corresponding suggestions for improvements:

• As clearly explained in the manuscript, haplotagging can be considered the reverse operation of read-based phasing. Thus, both operations are tightly linked, which is also the reason, why there are to date, to my knowledge, no tools dedicated solely to haplotagging. Typically, a phased VCF is the result from long read-based phasing (in the most accurate setting). Why would additional long-read data to be haplotagged not be used to improve the phased variant calls or to generate phased variant calls? Newly called and phased variants could simply be compared to the phased variants obtained elsewhere, or sequencing data combined to improve variant phasing. This is the common setting addressed by whatshap. Along these lines: when considering haplotagging errors in practice, one would need to consider incorrectly phased variants as source of haplotagging error. Can you please address these points in the manuscript?

• The manuscript’s title states that LRPhase is efficient and configurable. However, efficiency is stated or quantified neither with respect to complexity nor with respect to practical runtime. Further, the tool version 1.1.2 I tested had (except for mapping-related parameters) only the option to specify FDR or log likelihood threshold, which can both be considered a ranking of haplotagged reads, something that whatshap also allows (even though the authors mention they couldn’t find information on how to interpret the whatshap score). Please address and compare runtimes, state precisely adjustable tool parameters and consider changing the manuscript title.

• There are two scoring modes mentioned, one of them simply maximizing the number of matching alleles. There are no results presented for this scoring and it is not discussed and put in relation to the newly described scoring. Is the statistical model considering error rates indeed better and in what settings?

• The scoring model is in my opinion the major contribution of this paper, but it is provided only in the supplement and not evaluated with respect to parameters or in comparison to “scoring mode 2”. W.r.t. the model, a sequencing read base error rate e_r[a] is defined. I think that including the base accuracy at the read’s variant position is very useful. However, it seems to not be used and only a global error rate epsilon is used that is neither estimated nor a tool parameter as far as I see. The effect of epsilon is not evaluated and epsilon cannot be adjusted in the LRphase version provided. Can you please clarify the notion behind the error rates, fix the corresponding notation and evaluate error rate effects?

• In the Supplement, some math notation is introduced, but not used (e.g. S), please fix this.

• The entire Supplement should be moved to the main manuscript and carefully checked for notational correctness.

• The model is nicely introduced in general terms for any ploidy in the Supplement, however, the main manuscript indicates that only a ploidy of 2 can be used (since it always refers to maternal and paternal haplotypes). Can the tool handle ploidy >2? If no polyploidy is implemented, this is a limitation that needs to be mentioned (whatshap can handle polyploidy!).

• Is the model indeed a new contribution? Similar ideas seem to have been applied in the context of haplotagging before, e.g. Pubmed ID 31873213. Relationship to previously published similar models in the context of haplotagging need to be worked out and stated.

• I find the naming of the tool, LRphase, misleading, since phasing is typically rather considered the phasing of variants, which this tool does not do, it only performs haplotagging. I think that it is important to reflect this in the tool name. What about LRtagging?

• The description of related work is insufficient and needs additional attention. Only two tools are mentioned, whatshap haplotag and MarginPhase. With relation to their model, it is stated that they “appear to be based on the hidden-Markov model described in 15”. However, working out and stating the exact difference between the approaches and models and how this affects performance is important.

• Whatshap seems according to Pubmed ID 36335496 very comprehensive and versatile. Thus, specific user-configurable parameters and output metadata reported by LRphase and not by Whatshap and how exactly they help in identifying haplotagging errors needs to be specified to support the proposed benefits of LRPhase. When I tested LRphase as explained here (https://github.com/Boyle-Lab/LRphase/tree/main/example_data ), (i) I couldn’t find a documentation of the BAM file tags added by LRPhase and their interpretation (except HP for haplotype), (ii) I couldn’t find command-line parameters to adjust model parameters such as error rate, (iii) no metadata output was produced that helps in investigating individual read haplotaggings. According to my investigation of the tool documentation and the tool provided at bioconda, the current version 1.1.2 does not provide beneficial additional information or additional adjustable parameters except FDR.

• Please motivate why there is only one tool used for comparison, whatshap, and not at least also the other tool mentioned, MarginPhase.

• No information is provided on runtime and there are no runtime comparisons with the other tool. This needs to be added.

• The overall approach is extensively explained in the caption of Figure 1. The approach should be rather more clearly explained within the main text and only for illustration shown in Figure 1.

• Can you please comprehensively document the tool parameters and output specifications on the tool’s website?

• Can you please comment on the reliability of the haplotagging if the number of haplotagged reads is small? If simply lowest-scoring reads across an analyzed dataset are assigned to “unphased” to control FDR, then this is very sensitive to the number of reads being phased.

• How is the average sequencing error rate estimated? How is the output interpreted? Some tool parameters are not documented in the help, e.g. the mandatory (?!) parameter -s. Can multiple samples be processed at once? Please improve the documentation with respect to these points.

• Can you please provide the scripts used to rescue the GIAB VCF file and to simulate the read data?

• The notation of manuscript (especially Fig. 1) and model description (now Supplement) should be harmonized

• Many math symbols don’t show in the PDF and I cannot check them. This should be fixed. You may consider switching to Latex, especially since the formulas in the Supplement render poorly and are partly inconsistent in font.

Reviewer #2: In this work, the auhtors present LRphase, user-friendly tool that haplotags long sequencing reads based on a multinomial model and existing phased variant lists.

The overall result of the manuscript is very promising. The manuscript is overall readable.

The topic is interesting and it is well explained as well as the work.

The paper presents some issues to be improved. First, the introduction should focus on the goal of the work that it is not clear. Furthermore, from the state of the art it is not possible to understand if there are other similar works.

The authors should add a related work section.

Have they tried other state-of-the-arts methods ? If so, how were the results? If not, what was the reason?

The authors should add a comparison section.

Finally, the discussion of the results should be improved.

About the conclusion, the author should describe what are the future applications of research and the practical advantages.

6. PLOS authors have the option to publish the peer review history of their article (what does this mean?). If published, this will include your full peer review and any attached files.

Reviewer #1: No

Reviewer #2: No

---

## [Author Response · Author response to Decision Letter 0]

15 Nov 2023

Response to reviewers attached as separate document.

---

## [Decision Letter · Decision Letter 1]

1 Dec 2023

PONE-D-23-26110R1HaplotagLR: an efficient and configurable utility for haplotagging long readsPLOS ONE

Dear Dr. Boyle,

Thank you for submitting your manuscript to PLOS ONE. After careful consideration, we feel that it has merit but does not fully meet PLOS ONE’s publication criteria as it currently stands. Therefore, we invite you to submit a revised version of the manuscript that addresses the points raised during the review process.

We look forward to receiving your revised manuscript.

Kind regards,

Junwen Wang, Ph.D.

Academic Editor

PLOS ONE

Journal Requirements:

Reviewers' comments:

Reviewer's Responses to Questions

**Comments to the Author**

1. If the authors have adequately addressed your comments raised in a previous round of review and you feel that this manuscript is now acceptable for publication, you may indicate that here to bypass the “Comments to the Author” section, enter your conflict of interest statement in the “Confidential to Editor” section, and submit your "Accept" recommendation.

Reviewer #1: (No Response)

2. Is the manuscript technically sound, and do the data support the conclusions?

Reviewer #1: Partly

3. Has the statistical analysis been performed appropriately and rigorously? 

Reviewer #1: Yes

4. Have the authors made all data underlying the findings in their manuscript fully available?

Reviewer #1: No

5. Is the manuscript presented in an intelligible fashion and written in standard English?

Reviewer #1: Yes

6. Review Comments to the Author

Reviewer #1: The authors have properly addressed most of my concerns and I believe the manuscript and tool documentation improved significantly.

Remaining points are:

• The presentation of the model (section scoring model) is still poor and the formulas in the manuscript and supplement are inconsistent, e.g. in the supplement a posterior probability beta is introduced. Which formulas are correct? Please don’t put the mathematical introduction of the problem in the supplement. I still think that all supplementary content (mathematical introduction of model background phase sets, VCF etc.) should be moved to the main manuscript. When doing so, please remove the i (relating to an individual read) which is now used in the manuscript formulas, it makes the formulas unnecessarily complicated.

• Typo at “Therefore, a haplotag based on variant l may be reduced to a random guess”

• The epsilons (?) in section” Haplotagging Error Model” are still rendering incorrectly

• The script is not available from the link provided in the manuscript: “The rescued and filtered VCF and scripts to produce it from the original are archived at https://zenodo.org/record/7819303#.ZDW6sexudzU”

• Can you please improve the github documentation of “-e EPSILON, --epsilon EPSILON” and “-c, --epsilon_from_quality_scores” and be more specific in the manuscript: There are three settings discussed: (i) global error rate provided by user, (ii) global error rate as mean phred-score deduced from error rate across the entire read, (iii) every base has an individual error rate, computed from the base’s PHRED score. I think (iii) is by far the most reasonable. However, I cannot follow, which setting is which and whether all three or only two are implemented in the tool.

• I was not able to install the tool via conda using the command from the github documentation due to unresolvable incompatibilities, please fix that. Installation with pip worked.

• The peak memory usage is for both compared tools negligibly small, which should be stated.

7. PLOS authors have the option to publish the peer review history of their article (what does this mean?). If published, this will include your full peer review and any attached files.

Reviewer #1: No

---

## [Author Response · Author response to Decision Letter 1]

15 Dec 2023

Response attached as separate file.

---

## [Editor Report · Decision Letter 2]

30 Jan 2024

HaplotagLR: an efficient and configurable utility for haplotagging long reads

PONE-D-23-26110R2

Dear Dr. Boyle,

We’re pleased to inform you that your manuscript has been judged scientifically suitable for publication and will be formally accepted for publication once it meets all outstanding technical requirements.

Kind regards,

Junwen Wang, Ph.D.

Academic Editor

PLOS ONE
---

## [Editor Report · Acceptance letter]

3 Mar 2024

PONE-D-23-26110R2 

PLOS ONE

Dear Dr. Boyle, 

I'm pleased to inform you that your manuscript has been deemed suitable for publication in PLOS ONE. Congratulations! Your manuscript is now being handed over to our production team.

Kind regards, 

on behalf of

Prof. Junwen Wang 

Academic Editor

PLOS ONE